# Properties of Very Broad Line MgII Radio-Loud and Radio-Quiet Quasars

**Avinanda Chakraborty** [1], **Anirban Bhattacharjee** [2] **and Suchetana Chatterjee** [1,*]

1    Department of Physics, Presidency University, Kolkata 700073, West Bengal, India; avinanda.rs@presiuniv.ac.in
2    Department of Biology, Geology and Physical Sciences, Sul Ross State University, East Highway 90, Alpine, TX 79832, USA; axb14ku@sulross.edu
*    Correspondence: suchetana.physics@presiuniv.ac.in

**Abstract:** We perform an analysis of the properties of radio-loud (RL) and radio-quiet (RQ) quasars with MgII broad emission line (i-band magnitude $\leq$ 19.1 and z $\leq$ 1.9), selected from the parent sample of SDSS DR7 catalogue. For sources with full-width half maxima (FWHM) greater than 15,000 km s$^{-1}$ (very broad line sample; VBL) we find the radio loud fraction (RLF) to be about 40%. To further investigate this result we compare the bolometric luminosity, optical continuum luminosity, black hole (BH) mass and Eddington ratios of our VBL sample of RL and RQ quasars. Our analysis shows that in our VBL sample space, RL quasars have higher luminosities and BH mass than RQ quasars. The similarity in the distribution of their covering fraction (CF) shows that there is no difference in dust distribution between VBL RL and RQ quasars and hence dust is not affecting our results. We also find that there is no correlation of RL quasar properties with optical continuum luminosity and BH mass.

**Keywords:** radio-loud quasars; radio-quiet quasars; very broad line; bolometric luminosity; optical continuum luminosity; black hole mass; Eddington ratio; radio luminosity; covering fraction; radio-loudness

**Key Contribution:** It is known in the literature that roughly 10% of the total quasar population is radio-loud (RL). In this study we show that for broad MgII emission line quasars, we do observe an increase in RL fraction with full width half maximum (FWHM). For FWHM greater than 15,000 km s$^{-1}$, the radio loud fraction is greater than 40%.

## 1. Introduction

It has been widely reported in the literature that only a small fraction of the quasar population exhibits powerful radio emission in the form of radio jets e.g., [1–8], despite its very first detection in the radio band [9]. The radio-to-optical flux density ratio for optically selected quasars, appears to be bimodal, suggesting that there could be two distinct populations of objects, namely radio-loud (RL) and radio-quiet (RQ) quasars [2,4,6,10–12]. The search for bimodality has spanned different parameter spaces of quasar properties, namely, black hole (BH) mass [13], origin of radio emission [14], accretion rates [15,16], host galaxy properties [15,17,18], and BH spins [19–21]. Some recent studies suggest that black holes might be systematically heavier in RL sources than those in RQs [22–27].

The formation and launching of radio jets is not well understood, but the Blandford–Znajek (BZ) mechanism [19] remains as the most plausible theoretical explanation. It has been observed in many studies, that a correlation exists between emission line luminosities originated from the disk, and radio luminosities of jets in RL quasars [28–35]. It has been further reported that there exists some correlation between the radio luminosity and BH mass in RL quasars [13,36–38] but the claim has been debated by other groups [39–42].

In this work, we have identified a sample of RL and RQ quasars with broad MgII emission lines from the parent catalog of Shen et al. [43]. Our analysis shows that the radio loud fraction (RLF) increases with full width half maximum (FWHM) for our sample [44]. To further investigate this feature we construct a subsample (Very Broad Sample; VBL hereafter, FWHM > 15,000 km s$^{-1}$) from the broad MgII sample and perform a thorough comparison of the different properties of RL and RQ quasars.

## 2. Materials and Methods

We now provide a brief description of the datasets and the analysis methods that we utilized in this work. Our parent sample came from the Sloan Digital Sky Survey SDSS [45] Data Release 7 [46] quasar catalog [43]. It consisted of 105,783 spectroscopically confirmed quasars brighter than Mi = −22.0. Ref. [43] cross matched the quasar catalog of SDSS DR7 with the Faint Images of the Radio Sky at Twenty-cm (FIRST) survey. The quasars having only one FIRST source within 5″ were classified as core-dominated radio sources and those having multiple FIRST sources within 30″ were classified as lobe dominated. These two categories were together compiled as RL quasars by [43]. Quasars with only one FIRST counterpart between 5″ and 30″ were classified as RQ quasars. We checked optical spectra of the VBL quasars from SDSS, and the quasars which had inconsistencies in their broad MgII lines were manually discarded from our sample. Among SDSS quasars, the total number of quasars that had confirmed FIRST counterparts was 99,182. Among these quasars, 9399 were RL quasars and 88,979 were RQ quasars. We applied a redshift cut of z ≤ 1.9 for our MgII sample which restricted our sample size. Moreover, to avoid biasing in the radio properties, we further applied a cut in the i band magnitude (i < 19.1). Our final MgII VBL sample consisted of 14 RL and 22 RQ quasars.

## 3. Results

In [44], we studied the distributions of bolometric luminosity, optical continuum luminosity at 3000 Å and virial BH masses of RL and RQ quasars in our VBL sample and found them to be slightly different from the parent sample of [43]. We have found in [44] that our VBL RL quasars have marginally higher luminosities and BH masses compared to RQ quasars while in the parent sample there is no difference of the said properties between RL and RQ quasars. In this work we further investigated the properties of RL and RQ quasars of our VBL sample to search for clues regarding the increasing RLF with FWHM.

### 3.1. Studying Properties of the Very Broad Line Sample

In the left panel of Figure 1, we show the variation of optical continuum luminosity with BH mass for our VBL sample. From the figure we can clearly see RLs on average had higher luminosity as well as higher BH mass. Additionally, we observed that the four most massive objects which were also the most luminous ones were RL (upper right corner). In the right panel, we show the same for Eddington ratios and covering fractions (CF). We computed the Eddington ratios using virial BH mass and bolometric luminosities. Both the bolometric luminosity and BH mass were taken from [43] catalogue. We estimated CFs from the ratio of the integrated flux at mid-IR (MIR) to optical flux, which is the fraction of the optical flux emitted by the accretion disk that is absorbed and re-emitted into the IR by the dusty torus e.g., [47]. From the figure we observe that there was no difference in the mean values of both Eddington ratios and CF between RL and RQ quasars for our VBL sample. It has been claimed in some studies that the torus is a smooth continuation of the broad line region BLR [48]. However, it is known that the torus causes anisotropic obscuration outside the BLR. CF is often used to quantify this obscuration. Hence, the similarity in the distribution of CFs reveals that our VBL sample was mostly unbiased from obscuration effect.

We performed a correlation analyses between bolometric luminosity, BH mass, and FWHM of our VBL RL and RQ quasars and Tables 1 and 2 list the correlation coefficients. We observed in our sample that bolometric luminosity was not correlated with FWHM for

both RL and RQ quasars however bolometric luminosity correlated with black hole mass for both RL and RQ quasars. Our analysis suggests that the correlation between bolometric luminosity and BH mass was stronger in RQ quasars compared to the RL ones. Lastly, we observed a modest correlation between BH mass and FWHM for our RL sample. We however wish to note that our results were based on small sample statistics, namely 14 RL and 22 RQ quasars.

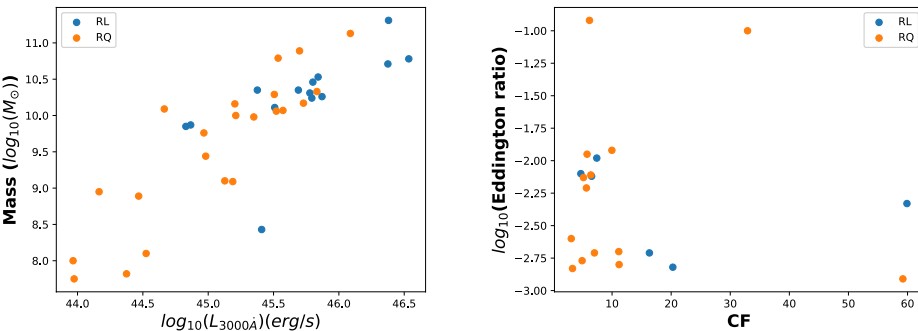

**Figure 1. Left panel**: Orange and blue solid circles denote the RQ and RL sources in our sample, respectively, on the optical continuum luminosity versus BH mass plane. **Right panel**: the same on the Eddington ratio versus covering fraction plane. From the figure we note that the Eddington ratio and covering fractions of the RL and RQ quasars in our sample are similar.

### 3.2. Studies of Radio Luminosities

We found that optical continuum luminosity and BH mass were higher for RL quasars in our VBL sample. We now focus on the radio luminosities of our RL sample. The left panel of Figure 2 shows variation of $L_{rad}$ (radio luminosity at 6 cm) with $L_{3000Å}$, optical continuum luminosity at 3000 Å and right panel showed the same with BH mass and $R$ [49], where $R$ is radio loudness (ratio of radio luminosity and optical continuum luminosity) for our VBL RL quasars. We did not find any significant dependence of $L_{rad}$ on $L_{3000Å}$ and $R$ on the BH mass.

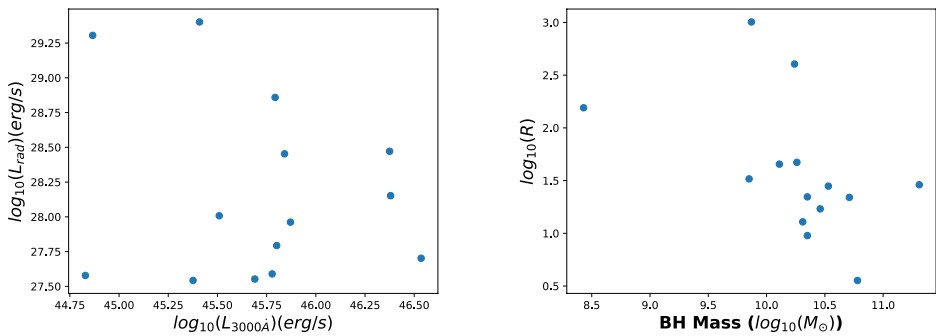

**Figure 2. Left panel**: The blue solid circles show the RL quasars in our sample on the optical continuum luminosity *versus* radio luminosity plane. **Right panel**: same on the $log_{10}(R)$ *versus* BH mass plane, where $R$ denotes the radio loudness. From the figure we note that there is no correlation between optical continuum luminosity and radio luminosity and between BH mass and radio-loudness.

**Table 1.** Correlation Matrix for RL quasars of MgII VBL Sample.

|  | $L_{Bol}$ | BH Mass | FWHM |
|---|---|---|---|
| $L_{Bol}$ | 1.0000 | 0.5922 | 0.2666 |
| BH mass | 0.5922 | 1.0000 | 0.5766 |
| FWHM | 0.2666 | 0.5766 | 1.0000 |

**Table 2.** Correlation Matrix for RQ quasars of MgII VBL Sample.

|          | $L_{Bol}$ | BH Mass | FWHM    |
|----------|-----------|---------|---------|
| $L_{Bol}$  | 1.0000    | 0.8713  | −0.0095 |
| BH mass  | 0.8713    | 1.0000  | 0.1304  |
| FWHM     | −0.0095   | 0.1304  | 1.0000  |

## 4. Summary

In this study, we show that the radio loud fraction is increasing with FWHM for MgII broad emission line quasars at z < 1.9. To investigate the cause for the high RLF we construct the VBL sample (FWHM > 15,000 km s$^{-1}$) and perform an analysis to compare the properties of RL and RQ quasars. From the optical continuum luminosity distributions, we observe that for VBL, RLs have higher mean luminosities implying RLs to be intrinsically brighter than RQs. The difference in BH mass is similar to that of bolometric luminosity.

We checked for the correlation between bolometric luminosity, BH mass, and FWHM for our VBL sample. We have observed that bolometric luminosity and BH mass are strongly correlated and RQs have a higher correlation compared to RLs. However bolometric luminosity and FWHM are uncorrelated for both the populations. We do not observe any correlation between BH mass and FWHM for RQs while there is modest correlation for the RL population. In future we plan to further investigate to establish the significance of our results, for example we plan to fit the discarded SDSS optical spectra and include them in our analysis. We further note that we need to check for beaming effects in our sample which we propose to undertake with future observations. Our results do not reveal any strong difference between the properties of our VBL RL and RQ quasars but provide a hint toward a higher BH mass and higher bolometric luminosity of RL quasars compared to RQs. We note that, this might have some implications on the disc and jet connections in RL quasars and provide us clues with possible differences in the disc structures of RL and RQ quasars. In future, we propose to undertake further investigations using other emission line diagnostics as well as better methods of BH mass estimation in quasar systems.

**Author Contributions:** Conceptualization, A.B. and A.C.; methodology, A.C. and A.B.; software, A.C.; validation, A.C., A.B. and S.C.; formal analysis, A.C.; investigation, A.C., A.B., S.C.; resources, A.C., A.B., S.C.; data curation, A.C.; writing—original draft preparation, A.C.; writing—review and editing, A.C., S.C., A.B.; visualization, A.C.; supervision, S.C.; project administration, S.C.; funding acquisition, S.C. All authors have read and agreed to the published version of the manuscript.

**Funding:** This research was funded by the SERB-ECR grant (grant number: ECR/2016/000005) and SERB-CRG grant (grant number: CRG/2020/002064) of Suchetana Chatterjee from the Department of Science and Technology. SC further acknowledges support from the Board of Research in Nuclear Sciences through the grant 57/14/10/2019-BRNS/34094.

**Institutional Review Board Statement:** Not applicable.

**Informed Consent Statement:** Not applicable.

**Data Availability Statement:** http://vizier.cfa.harvard.edu/viz-bin/VizieR?-source=J/ApJS/194/45 (accessed on 29 September 2021).

**Acknowledgments:** A.C. thanks Mike Brotherton and Jaya Maithili from the University of Wyoming, Ritaban Chatterjee from the Presidency University, Swamtrupta Panda from the Nicolaus Copernicus Astronomical Center and Preeti Kharb from the National Centre for Radio Astrophysics, for their valuable inputs. S.C. is grateful to the Inter University Center for Astronomy and Astrophysics (IUCAA) for providing infra-structural and financial support along with local hospitality through the IUCAA-associateship program.

**Conflicts of Interest:** The authors declare no conflict of interest.

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
