# Peer review of "Properties of Very Broad Line MgII Radio-Loud and Radio-Quiet Quasars"

_galaxies, doi:10.3390/galaxies9040074_

Round 1
Reviewer 1 Report
General comments:
- Abstract. I would avoid referring directly to the publication in the Abstract. I suggest entering the catalog name here.
- Abstract. I think that the first sentence wrongly presents the meaning of the analyzes presented in the article. I think that the authors present ‘the analysis of the properties of RL and RQ quasars with broad emission line MgII” and not ‘the analysis of the broad emission lines of RL and RQ quasars’? I’m confused here.
- Abstract and Chapter 2. I understand that (FWHM > 15,000 km s^−1) is one of the sample selection criteria but it is not covered in Chapter 2 where this selection is described. At what point in the sample selection process is this criterion applied?
- Chapter 3.1. Here again, the chapter title is confusing. The authors discuss in it the properties of the broad-line quasars and not the properties of the MgII line?
- What is the difference between chapters 3.1 and 3.2? They both describe the correlation studies in my opinion. I think these results should be better organized and the distinction should be different. Maybe Chapter 3.1 should describe the correlations between RL and RQ quasars and chapter 3.2 properties of RL quasars.
- Chapter 4. This is not actually a conclusion but a summary. I am missing here any interpretation of these obtained results, i.e. what these positive and what these negative correlations mean. There should be at least some preliminary interpretation.
Minor comments:
- Line 69 – I think the word ‘there’ is unnecessary here.
- Line 91 – ‘Left panel of 2’ should be ‘Left panel of Figure 2’
- Line 96 – ‘and 1 and 2 list’ should be ‘and Tables 1 and 2’.
- Line 102 - Missing dot on the end of a sentence.
My general impression of this work is that it is written quite chaotically. I think the authors should better think through the layout of the work to make it easier to read, especially when it comes to presenting the results.
I also think that the work should be read by a language editor because some sentences seem clunky to me and you have to guess what they are about or go back to the previous sentences.
In general, the very idea of research seems interesting to me, but it should be better presented.
Reviewer 2 Report
This conference paper entitled “Properties of Very Broad Line MgII Radio-loud and Radio-quiet Quasars” performed a systematic study of radio-loud and radio-quite quasars (RL/RQ) by way of emission line of MgII. Actually, the classification on loud/quiet quasars is under debate but this work has shed new light on this topic. Besides, some physical parameters, such as the black hole mass, bolometric luminosity, optical luminosity and etc, have been discussed, and the author obtains some results that are consistent with other works. I believe this paper has been merited the publication in this proceedings. I also provide some comments as follows and hope the author can take them into account.
General concerns:
- line 76, page 2. “We compute the Eddington ratios using virial BH mass and bolometric luminosity.” I am wondering if the mass and bolometric luminosity come from the author’s previous work (AN, 2021, 342, 142)? Or collected from other work? I suggest it should be claimed clearly here.
- line 93, page 3. “...where R is radio loudness (...).” Please cite this useful reference Zhang et al. (2021), PASJ, 73, 313, in which the author studies the RL/RQ boundary using optical and radio data of AGNs based on the Bayesian analysis technique, and obtains a divide between two subclasses is logR=1.26.
Minor comments:
- line 71, page 2. “In Figure 1 left panel”-->”In the left panel of Figure 1”
- line 93, page 3. “...and 1 and 2 list the correlation”--> “...and Table 1 and 2 list the correlation”
